# Impact of COVID-19 lockdown on psychosocial factors, health, and lifestyle in Scottish octogenarians: The Lothian Birth Cohort 1936 study

**Adele M. Taylor[1]**, **Danielle Page[1]\***, **Judith A. Okely[1]**, **Janie Corley[1]**, **Miles Welstead[1]**, **Barbora Skarabela[1]**, **Paul Redmond[1]**, **Tom C. Russ[1,2,3]**, **Simon R. Cox[1]**

1 Lothian Birth Cohort Group, Department of Psychology, University of Edinburgh, Edinburgh, United Kingdom, 2 Alzheimer Scotland Dementia Research Centre, University of Edinburgh, Edinburgh, United Kingdom, 3 Division of Psychiatry, Centre for Clinical Brain Sciences, University of Edinburgh, Edinburgh, United Kingdom

☉ These authors contributed equally to this work.

\* danielle.page@ed.ac.uk

**Data Availability Statement:** Data cannot be shared publicly because of the data containing information that could compromise participant

## Abstract

### Background

Little is known about effects of COVID-19 lockdown on psychosocial factors, health and lifestyle in older adults, particularly those aged over 80 years, despite the risks posed by COVID-19 to this age group.

### Methods

Lothian Birth Cohort 1936 members, residing mostly in Edinburgh and the surrounding Lothians regions in Scotland, mean age 84 years (SD = 0.3), responded to an online questionnaire in May 2020 ($n$ = 190). We examined responses (experience and knowledge of COVID-19; adherence to guidance; impact on day-to-day living; social contact; self-reported physical and mental health; loneliness; and lifestyle) and relationships between previously-measured characteristics and questionnaire outcomes.

### Results

Four respondents experienced COVID-19; most had good COVID-19 knowledge (94.7%) and found guidance easy to understand (86.3%). There were modest declines in self-reported physical and mental health, and 48.2% did less physical activity. In multivariable regression models, adherence to guidance by leaving the house less often associated with less professional occupational class (OR = 0.71, 95%CI 0.51–0.98) and poorer self-rated general health (OR = 0.62, 95%CI 0.42–0.92). Increased internet use associated with female sex (OR = 2.32, 95%CI 1.12–4.86) and higher general cognitive ability (OR = 1.53, 95%CI 1.03–2.33). Loneliness associated with living alone (OR = 0.15, 95%CI 0.07–0.31) and greater anxiety symptoms (OR = 1.76, 95%CI 0.45–1.24). COVID-19 related stress associated with lower emotional stability scores (OR = 0.40, 95%CI 0.24–0.62). Decreased

consent and confidentiality. Data are available from the Lothian Birth Cohort team, University of Edinburgh (contact via lbc1936@ed.ac.uk) for researchers who meet the criteria for access to confidential data. Full details of the data access procedure are available here at (https://www.ed.ac.uk/lothian-birth-cohorts/data-access-collaboration) alongside the requisite forms and data dictionaries.

**Funding:** This research was conducted by the LBC1936 study team, which is funded by Age UK (Disconnected Mind programme grant). Additional funding from the UK Medical Research Council (MRC; G0701120, G1001245, MR/M013111/1), the National Institutes of Health (NIH; R01AG054628) and the University of Edinburgh is gratefully acknowledged. Age UK review overall plans for the LBC1936 study as part of the peer-review process during grant application. The funders had no role in study design, data collection and analysis, decision to publish, or preparation of the manuscript.

**Competing interests:** The authors have declared that no competing interests exist.

physical activity associated with less professional occupational class (OR = 1.43, 95%CI 1.04–1.96), and lower general cognitive ability (OR = 0.679, 95%CI 0.491–0.931).

## Conclusions

Characteristics including cognitive function, occupational class, self-rated health, anxiety, and emotional stability, may be related to risk of poorer lockdown-related psychosocial and physical outcomes.

## Introduction

Since the declaration of pandemic on 11th March 2020 [1], public health measures have been implemented across the globe to suppress the spread of the coronavirus disease 2019 (COVID-19). In Scotland, lockdown measures introduced on 23rd March 2020 included social and physical distancing, isolation of symptomatic individuals, and restrictions on leaving the home (once daily for essential reasons) [2]. The effects of COVID-19 lockdown measures on older people are yet to be determined, especially in those over the age of 80, who are classed as a 'high risk' group in Scotland [3]. The current study aimed to examine the impact of the Scottish COVID-19 lockdown on psychosocial factors, health, and lifestyle in older adults aged approximately 84-years from the Lothian Birth Cohort 1936 (LBC1936) study.

In Scotland, older people are considered to be at higher risk of severe illness from COVID-19 [3–6]. He risk increases for individuals with chronic comorbidities, particularly ageing-related diseases including cardiovascular disease, diabetes, respiratory and chronic pulmonary disease [5, 6]. Over half of those aged over 80 are estimated to be at high risk due to underlying health conditions [7]. Older people are also at higher risk of mortality; in Scotland, 77% of all deaths involving COVID-19 to 14th June 2020 were of people aged 75 and over [8]. A prospective cohort study of individuals aged between 0 and 104 (median age 73) in UK acute care hospitals found the highest proportion of hospitalisations and mortality among those aged 80 and over [5]. Because of their increased risk, those in the 'high risk' category when lockdown began in Scotland were asked to 'shield', remaining at home and strictly avoiding social contact with anyone outside of their homes for at least 12 weeks.

Effects of lockdown measures on 'high risk' individuals who remain illness-free are unclear. Social distancing measures inherently limit activities and promote social isolation, potentially to the detriment of physical and mental health [9]. In middle-aged and older adults, isolation and loneliness are associated with poor cognitive function, cognitive decline, depression, anxiety, lack of feeling valued, poor physical health including poor cardiovascular function, immunity, and mortality [10–16]. There are physical health risks associated with reduced activity during lockdown [17], which warrant consideration given the association between declines in physical fitness and cognitive function [18]. Prior research indicates the relationship between physical health and psychosocial risk factors is relevant for the COVID-19 pandemic [19, 20]. Recent research examining the same data as in the current study found that levels of physical activity significantly decline during lockdown, and higher levels of physical activity during lockdown was associated with greater extraversion and a higher age 11 IQ [21]. Labelling older people as a homogenous group of vulnerable individuals may result in stereotyping or marginalisation [22], and negative consequences of social isolation may be exacerbated by the 'digital divide' [9], since older people may disproportionately face barriers to accessing modern technology and information sources. That said, it is possible that many older people are more resilient than commonly portrayed, and have adequate resources to cope well.

Data from older people during the pandemic are surprisingly limited, given that they are 'high risk' and any effects of strict lockdown in this group are not clear. The current study is particularly concerned with those over the age of 80, who are under-represented in the COVID-19 literature. No study to date has focused solely on adults over the age of 80. Even in high-quality large-scale studies of COVID-19 with representative samples of hundreds or thousands of participants, the number of individuals sampled over the age of 80 is low [23–25], with conclusions drawn from sample sizes fewer than as low as 20 [26], and in some studies those over 80 are excluded entirely [27]. Given the clear risks of the virus to older adults, both in terms of health and the wider psychological, social, and lifestyle impacts resulting from stringent lockdown measures, it is important that the experiences of older adults are well reported. Findings from the general population and past pandemics suggest negative consequences for older people in terms of anxiety and depression [28, 29], psychological distress [30–33], and wellbeing [34]. At the beginning of lockdown, survey participants from across Great Britain rated social isolation and practical concerns as being of greater risk to their mental health and wellbeing than fear of contracting COVID-19 [26]. Individuals aged over 75 were more likely to report high anxiety than any other age group [35]. Physical health may be adversely affected due to the impact of lockdown on behaviours such as sleep [36] and physical activity [37]. Furthermore, the experience is likely to vary between individuals based on sociodemographic differences [38–40], physical ability [40], genetics [41], mood and personality [28]. One of few studies to report on mostly middle aged and older adults (aged 23–88, with 63% of the sample over age 60) found differences in COVID-19 knowledge, awareness, attitudes, and behaviours across ethnic and socioeconomic groups, and in relation to differing levels of health literacy [23]; being unemployed or retired, having poorer health, and having lower health literacy were associated with poorer COVID-19 knowledge and fewer changes to daily routine.

To fully understand the impact of Scotland's lockdown measures on older people, and inform future interventions in the event of a 'second wave' or other health crises, it is important to measure: the ways in which behaviours and routines have been altered; how physical and mental health have been affected; whether some people have fared better than others; and whether there are risk and protective factors associated with these differences. Existing research cohorts are particularly valuable in understanding the impacts of the COVID-19 pandemic, particularly by 'embedding research on COVID-19 into studies where participants' mental or cognitive health has previously been ascertained' [9]; this is a key strength of the current study. This study is one of few with a reasonably sized sample of older adults; many others base their findings on the responses of very few older-age participants. We explored the impact of lockdown measures on community-dwelling older adults from the LBC1936 study by linking responses to a COVID-19 questionnaire at age 84 with rich data on cognitive ability, demographics, psychosocial, and health factors previously collected at age 82. The study had two aims. First, to describe responses to the COVID-19 questionnaire. Second, to use bivariate and multivariate analyses to examine relationships between previously collected participant characteristics and psychosocial factors, health and lifestyle during lockdown.

## Methods

### Participants

Participants were members of the LBC1936 study, a longitudinal study principally investigating non-pathological cognitive and brain ageing. All 1,091 members were born in 1936; most reside in Edinburgh and the surrounding Lothian region of Scotland and took part in the Scottish Mental Survey 1947 (SMS1947) [42]. Participants were recruited between 2004 and 2007

at mean age 70 years (wave 1) [43]. To date, they have attended four further waves at mean ages 73 (2007–2010, n = 866), 76 (2011–2013, n = 697), 79 (2014–2017, n = 550), and 82 (2017–2019, n = 431). At each wave, detailed cognitive ability, health, psychosocial, lifestyle, and other data are collected. Information on tracing, recruitment and testing of LBC1936 participants can be found elsewhere [44, 45]. The current study is based on a subsample of participants (n = 190) who completed an online COVID-19 questionnaire at mean age 84 (± 0.3) years; this group is referred to as 'respondents'. Ethical approval for the Lothian Birth Cohort 1936 study was obtained from Multi-Centre Research Ethics Committee for Scotland (MREC/01/0/56; Wave 1), the Lothian Research Ethics Committee (LREC/2003/2/29; Wave 1), and the Scotland A Research Ethics Committee (07/MRE00/58; Waves 2–5). Ethical approval for the LBC1936 COVID questionnaire described in the current study was granted as an amendment to 07/MRE00/58 (AM18). The study complies with Declaration of Helsinki guidelines.

## LBC1936 COVID-19 questionnaire

All LBC1936 participants registered with the study in May 2020 (n = 454) were invited by letter to take part in an online COVID-19 questionnaire, designed by the LBC1936 team for this study. A summary of questionnaire items and measurement can be found in Table 1; for questionnaire in its entirety, see S1 Appendix. The LBC1936 are a volunteer sample; inclusion criteria for the present study were defined as any participant who had completed at least one wave of testing, and who had the means to access and complete the questionnaire online by themselves or with assistance. Participants gave full and informed written consent by completing an online consent form before taking part in the questionnaire. Respondents lacking capacity to provide informed consent or unable to complete the questionnaire themselves (n = 3) were permitted to have assistance (e.g. from guardian or nearest relative). Instructions were included in the participant information sheet stating that for any participant requiring assistance with completing the online questionnaire, the person providing assistance should contact the LBC study team before beginning the questionnaire to confirm their status as a Welfare Power of Attorney, Guardian, or Nearest Relative. The LBC study medic specialising in geriatric medicine advised the team on whether this individual was suitable for providing consent on behalf of any participant lacking capacity to provide informed consent. The questionnaire was built using the Qualtrics XM platform, and was live between May 27th and June 8th 2020. The questionnaire took approximately 30 minutes to complete; it consisted of 145 questions examining experience of COVID-19, knowledge and adherence to guidance, impact on day-to-day living, social contact, self-reported physical and mental health, loneliness, and lifestyle factors. Questions were designed to harmonise with existing COVID-19 surveys including Generation Scotland's CovidLife Survey [46], the Chicago COVID-19 Comorbidities (C3) Survey [47], and surveys from other studies of COVID-19 attitudes and behaviours [23]. Many questions were adapted from these surveys and had Likert-type response scales [23, 46]; all were optional. Some questions refer to the period 'since COVID-19 measures were introduced on 23rd March 2020', hereinafter referred to as 'lockdown'.

## Measures

**Questionnaire measures.** We examined responses to the COVID-19 questionnaire (experience of COVID-19; knowledge and adherence to guidance; impact on day-to-day living; social contact; self-reported physical and mental health and loneliness; and lifestyle (see S1–S6 Tables for the wording of individual items and response options).

**Table 1. Description of COVID-19 questionnaire outcome measures and covariates used in correlation and regression analyses.**

| Measure | Item wording/measurement method | Responses |
|---|---|---|
| Questionnaire outcome measures (at age 84) | | |
| Adherence to guidance | | |
| Decreased frequency of leaving home | 'How often have you been leaving your home since COVID-19 measures were introduced (23rd March 2020)?' | More than once per day/once per day or less |
| Impact on day-to-day living | | |
| Increased internet usage | 'How has your internet usage changed since COVID-19 measures were introduced (23rd March 2020)?' | More internet use/same or less internet use |
| Gets additional help | 'Have you received any additional help in your daily life with things such as grocery shopping, errands, or picking up medications since COVID-19 measures were introduced?' | Yes/No |
| Greater change in daily routine | 'How much has COVID-19 changed your daily routine?' | A lot/somewhat/a little/not at all |
| Self-reported physical and mental health and loneliness | | |
| Greater COVID-19-related stress or nervousness | 'In the last two weeks, how often have you felt nervous or stressed because of COVID-19?' | Sometimes/never |
| Poorer self-reported physical health | 'In general, since the COVID-19 measures were introduced, would you say your physical health is:' Adapted from the Short Form 36 (SF-36) [48]. | Excellent/very good/good/fair/poor |
| Poorer self-reported mental health | 'In general, since the COVID-19 measures were introduced, would you say your emotional and mental health is:' Adapted from the SF-36 [48]. | Excellent/very good/good/fair/poor |
| Experiencing Loneliness | 'How often have you felt lonely during the past week?' | Sometimes/never |
| Lifestyle | | |
| Decrease in physical activity | 'Compared to before COVID-19 measures were introduced (23rd March 2020), how much physical activity are you doing now? This includes activities that make you breathe harder than normal (e.g., brisk walking).' | Much more/slightly more/the same/ slightly less/much less |
| Returning to or taking up a new pastime | 'Since COVID-19 measures have been in place (23rd March 2020), have you returned to or started up a new pastime that you can do from home?' | Yes/No |
| Covariates | | |
| Demographic | | |
| Childhood occupational class* | Father's highest obtained occupation reported at wave 1 (mean age 70); scored according to General Register Office's Census 1951 Classification of Occupations [49]. | 1 (professional)– 5 (unskilled) |
| Adulthood occupational class | Participant's highest occupation reported at wave 1 (mean age 70); scored according to Office of Population Censuses and Surveys' Classification of Occupations, 1980 [50]. | 1 (professional)– 5 (unskilled) |
| Age | Age in days at time of questionnaire (mean age 84) or wave 5 (mean age 82). | |
| Sex | Collected at wave 1 (mean age 70). | Male/female |
| Years of education* | Self-reported years of full-time education' reported at wave 1 (mean age 70). | |
| Marital status* | Marital status reported at time of questionnaire (mean age 84) or wave 5* (mean age 82). | Married/Other |
| Living alone | Living status reported at time of questionnaire (mean age 84) and at wave 5* (mean age 82) | Yes/No |
| Area of residence* | Current area of residence reported at time of questionnaire (mean age 84). | Rural/Urban/Suburban |
| Cognitive | | |
| Age 11 cognitive ability* | Moray House Test No.12 (MHT) scores from the SMS1947 (39). The MHT is well-validated test of general cognitive ability which includes questions on: following directions, same-opposites, word classification, analogies, practical items, reasoning, proverbs, arithmetic, spatial items, mixed sentences, and cypher decoding. | Sum of correct items out of total of 76† |
| Mini-mental state examination score* | Mini-mental state examination (MMSE) [51] scores at wave 5 (age 82). The MMSE is a widely-used screening instrument for cognitive function, which assesses basic ability in language, recall memory, attention, and orientation to time and place. MMSE scores, which range from 0 to 30, can been used to assess the presence of cognitive impairment. Scores of less than 24 are commonly considered to indicate possible mild cognitive impairment or dementia. | Sum of correct items out of total of 30 |

(*Continued*)

**Table 1.** (Continued)

| Measure | Item wording/measurement method | Responses |
|---|---|---|
| Fluid cognitive ability 'gf' | Score derived from principal component analysis (PCA) of scores on six subtests from the Weschler Adult Intelligence Scale [52, 53] designed to assess fluid-type abilities at wave 5 (age 82; Matrix reasoning, Block Design, Digit symbol coding, Digit Span Backwards, Letter-number sequencing, and Symbol Search). Fluid-type abilities are those related to processing and integration of information and novel problem solving that do no rely on learned or acquired knowledge. | |
| General health literacy | Score derived from a PCA of age-73 (wave 2) scores on three functional health literacy measures (Rapid Estimate of Adult Literacy in Medicine; Shortened Test of Functional Health Literacy in Adults; Newest Vital Sign) [54]. Health literacy is the capacity to acquire, process and use health information to successfully navigate all aspects of health, including the ability to use health documents, interact with healthcare professionals and undertake health-promoting behaviours to prevent future ill health. | |
| **Health and physical fitness** | | |
| Number of chronic comorbidities | Sum of conditions based on self-reported history of cardiovascular disease, hypertension, and diabetes at wave 5 (age 82). | All items: yes/no; total ranges from 0 to 3 |
| Undiagnosed diabetes* | Blood glycated haemoglobin at wave 5 (age 82; HbA1c; IFCC units). | |
| Lung function* | Forced expiratory volume in 1s at wave 5 (age 82; FEV1). | |
| Grip strength* | Best overall grip strength performance of three attempts each in right and left hands at wave 5 (age 82; kg). | |
| Townsend Disability Scale score [55] | Townsend Disability Scale score at wave 5 (age 82). | Sum of responses out of total of 18 (higher scores indicate poorer ability). |
| Body Mass Index (BMI)* | Weight in kg (electronic SECA scales with digital display) divided by squared height in metres, (SECA stadiometer) measured by research nurse at wave 5 (age 82; kg/m2). | |
| Self-rated general health | At wave 5 (age 82): 'In general, would you say your health is:' Question from the SF-36 [48]. | Excellent/very good/good/fair/poor |
| **Mood** | | |
| Anxiety symptoms | Summed anxiety item scores from Hospital Anxiety and Depression scale [56] at wave 5 (age 82). | Sum of 7 item scores, scored 0–3; total score out of 21. |
| Depression symptoms | Summed depression item scores from Hospital Anxiety and Depression scale [56] at wave 5 (age 82). | Sum of 7 item scores, scored 0–3; total score out of 21. |
| **Personality** | | |
| Emotional stability | Measured using the 50-item IPIP Big-Five personality inventory [57] at wave 5 (age 82). | Sum of 10 items scored 1–5; total score out of 50. |
| Extraversion | Measured using the 50-item IPIP Big-Five personality inventory [57] at wave 5 (age 82). | Sum of 10 items scored 1–5; total score out of 50. |
| Conscientiousness | Measured using the 50-item IPIP Big-Five personality inventory [57] at wave 5 (age 82). | Sum of 10 items scored 1–5; total score out of 50. |

*Covariate measure used only for comparison of respondents versus non-responders; not included in further analysis.

**Covariates.** Measures hypothesised to be associated with COVID-19 questionnaire outcomes were selected a priori based on previous associations between these variables and psychosocial factors, health and lifestyle [54, 58–68]. These included: childhood and adulthood occupational social class; age; sex; years of formal full-time education; marital status; living alone; current area of residence; age-11 cognitive ability; Mini-Mental State Examination score [51]; fluid cognitive ability 'gf'; general healthy literacy; chronic comorbidities; undiagnosed diabetes; lung function; grip strength; Townsend Disability Scale Score [55]; Body Mass Index (BMI); self-rated general health; emotional stability; extraversion; and conscientiousness. Measurement is described in Table 1. Additional covariates were selected solely for analysis of characteristics of those who responded to the questionnaire compared to non-responders (see Table 1).

## Statistical analysis

Statistical analyses were conducted using R v3.6.3 [69] and IBM SPSS Statistics v.25 [70]. Results of the PCA for covariates gf and general health literacy are presented in S7 Table. Descriptive statistics for questionnaire responses were percentages of response relative to number of respondents per questionnaire item (S1–S6 Tables). An alpha level of .05 was employed for all statistical tests. Welch's 2-sample t-test, chi-squared tests with Yates' continuity correction, and Fisher's exact test were used to compare characteristics of respondents versus non-responders (Table 2). Before undertaking further analysis, respondents who did not

**Table 2. Comparison of background characteristics for Lothian Birth Cohort 1936 participants who responded to the COVID-19 questionnaire versus those who did not respond.**

| Background characteristic | Respondent (N = 190) | | Non-responder (N = 264) | | Difference test | | |
|---|---|---|---|---|---|---|---|
| | Mean/N | SD/% | Mean/N | SD/% | t/$\chi^2$ | p | Cohen's d |
| **Age (years)*** | 81.98 | 0.46 | 82.02 | 0.48 | 0.96 | 0.34 | 0.09 |
| **Sex (n male)** | 96 | 52.7 | 113 | 45.4 | 2.00 | 0.16 | - |
| **Childhood occupational class** | 2.83 | 0.94 | 2.88 | 0.96 | 0.46 | 0.65 | 0.05 |
| **Adulthood occupational class** | 2.00 | 0.83 | 2.40 | 0.92 | 4.70 | < .001 | 0.46 |
| **Years of formal full-time education** | 11.21 | 1.17 | 10.7 | 1.12 | -4.83 | < .001 | 0.47 |
| **Marital status*** | | | | | 3.41 | 0.06 | - |
| **Married** | 110 | 60.4 | 127 | 51.0 | | | |
| **Not married** | 72 | 39.6 | 122 | 49.0 | | | |
| **Living alone* (n yes)** | 60 | 33.0 | 114 | 45.8 | 6.65 | 0.01 | - |
| **Moray House test (MHT) score at mean age 11†** | 53.33 | 10.63 | 48.65 | 12.30 | -5.99 | < .001 | 0.48 |
| **Mini mental state examination (MMSE) score*** | 28.77 | 1.82 | 27.85 | 2.55 | -4.40 | < .001 | 0.42 |
| **General cognitive ability*** | 0.37 | 0.92 | -0.28 | 0.97 | -6.78 | < .001 | 0.69 |
| **General health literacy score at mean age 73** | 0.45 | 0.82 | -0.08 | 0.97 | -5.71 | < .001 | 0.59 |
| **Body Mass Index (BMI)*** | 26.95 | 3.92 | 27.29 | 4.43 | 0.84 | 0.40 | 0.08 |
| **Grip strength (kg; max in dominant hand)*** | 27.82 | 8.77 | 25.77 | 8.62 | -2.37 | 0.02 | 0.24 |
| **Forced expiratory volume in 1s (FEV1)*** | 2.12 | 0.64 | 1.97 | 0.61 | -2.51 | 0.01 | 0.25 |
| **History of hypertension history (n yes)*** | 110 | 60.4 | 138 | 56.6 | 0.50 | 0.48 | - |
| **History of cardiovascular disease (n yes)*** | 75 | 41.4 | 95 | 38.3 | 0.31 | 0.58 | - |
| **History of diabetes (n yes)*** | 19 | 10.4 | 32 | 12.9 | 0.38 | 0.54 | - |
| **Glycated haemoglobin (HbA1c)*** | 40.02 | 7.44 | 40.58 | 8.29 | 0.71 | 0.48 | 0.07 |
| **Townsend disability scale score*** | 1.35 | 2.20 | 2.36 | 3.45 | 3.67 | < .001 | 0.35 |
| **Self-reported health*** | | | | | 13.61 | 0.008 | - |
| Excellent | 23 | 12.6 | 19 | 7.6 | | | |
| Very good | 86 | 47.3 | 91 | 36.5 | | | |
| Good | 61 | 33.5 | 102 | 41.0 | | | |
| Fair | 10 | 5.5 | 29 | 11.6 | | | |
| Poor | 2 | 1.1 | 8 | 3.2 | | | |
| **Anxiety symptoms*** | 3.66 | 2.96 | 4.53 | 2.94 | 3.01 | 0.003 | 0.29 |
| **Depression symptoms*** | 2.72 | 2.22 | 3.43 | 2.50 | 3.09 | 0.002 | 0.30 |
| **Emotional stability *** | 37.07 | 7.21 | 34.64 | 6.47 | -3.54 | < .001 | 0.36 |
| **Extraversion*** | 32.63 | 7.41 | 30.64 | 7.07 | -2.75 | 0.006 | 0.28 |
| **Conscientiousness*** | 38.50 | 5.74 | 36.74 | 6.13 | -2.92 | 0.003 | 0.30 |

For continuous variables, p-values are for differences calculated using Welch's 2-sample t-test for continuous variables with Cohen's d for standardised effect size. For categorical variables, p-values are for Fisher's exact test or $\chi^2$ test with Yates's continuity correction.

*Measures were recorded at the most recent full wave of data collection when participants were mean age 82 years old.

† Difference test results for Moray House Test scores at mean age 11 are based on age-adjusted values.

attend the most recent wave of LBC1936 testing (wave 5; $n$ = 8) were excluded, leaving an analytic sample of 182 for inclusion in correlations and regression models. Some outcome measures were recoded from categorical to binary due to low numbers in some response categories; details of outcome measures for correlations and regressions, some of which were recoded, are included in Table 1. For raw response frequencies see S1–S6 Tables.

We conducted exploratory bivariate Spearman's rank correlations to identify relationships between previously measured characteristics and COVID-19 questionnaire outcomes from the subthemes: adherence to guidance, impact on day-to-day living, self-reported physical and mental health and loneliness, and lifestyle. We report significant correlations after adjustment for multiple comparisons using Holm-Bonferroni correction [71]. Variables that were significantly correlated with COVID-19 questionnaire outcomes were included in binary or ordinal logistic regression models to examine their relative importance and to adjust for potential confounding. All models were adjusted for age and sex. Additional covariates for each model were selected on the basis of significant correlations with COVID-19 questionnaire outcomes. These were entered consecutively into regression models based on variable subtype in the following order: age and sex, demographics, cognitive ability, health, mood, personality. We report odds ratios (OR) and confidence intervals (CI) for significant associations in final models after adjustment for all covariates. Associations with p-values < .005 remained significant after correction for multiple testing using false discovery rate (FDR) correction [72]. Odds ratios reported for continuous independent variables relate to a 1SD increase.

In an additional exploratory step, we conducted Wilcoxon signed rank tests to test for significant changes between 'before' and 'during' lockdown ratings for self-reported physical and mental health (reported as part of the online questionnaire). We derived physical and mental health change scores by subtracting 'before' from 'during' scores, then examined possible correlations with previously measured characteristics to explore potential predictors of change.

## Results

### Comparison of responders and non-responders

Background characteristics of respondents ($n$ = 190) and non-responders ($n$ = 264) are presented in Table 2. Respondents were less likely to live alone and tended to have had a more professional occupational status; more years of formal education; higher cognitive ability scores; better physical fitness and self-rated general health; fewer symptoms of anxiety and depression; and higher scores for personality traits emotional stability, extraversion, and conscientiousness (all $p$-values ≤.02; Cohen's d: 0.25 to 0.69).

### Questionnaire responses

**Experience of COVID-19.** Of 190 respondents, 4 (2.1%) reported a self-diagnosis of COVID-19 based on symptoms (see S1 Fig); 13.7% were advised to shield due to an underlying health condition; and 12.6% postponed contacting a medical service or attending a medical appointment due to anxiety about COVID-19 (S1 Table).

**Knowledge and adherence to guidance.** The majority (94.7%) rated their COVID-19 knowledge extremely or somewhat good, and 86.3% found Scottish Government COVID-19 guidance extremely or somewhat easy to understand. Almost all followed guidance in relation to leaving the home once daily or less (97.9%), social distancing (98.9%), staying at home (96.8%), hand-washing (97.9%), and self-isolating if suffering COVID-19 symptoms (88.6%) all or most of the time. 70.5% said they were unlikely to accidentally come into close contact with someone not in their household (i.e. less than 2 metres) when leaving their home

(S2 Table). Most (94.1%) followed COVID-19-related news daily; the BBC was the most frequently used source and was rated most helpful (S2 and S3 Figs).

**Living situation and impact on day-to-day living.** Over one-third of respondents (38.4%) were living alone and 56.3% were living with a partner during lockdown. 60.0% lived in a suburban area, and almost all had access to a shared or private garden (91.9%). Almost three-quarters (73.8%) reported change in their daily routine during lockdown. Nearly two-thirds (62.6%) received help from others during lockdown, and 64.2% changed their prescription or method of ordering in order to continue to access prescribed medicines during lockdown. Half of respondents were aware of local initiatives to help those self-isolating (51.6%), whereas 42.1% did not know. Nearly two-thirds used more non-cash alternatives during lockdown, and 35.3% said using cash was important. 54.5% used the internet more often during lockdown and 37.1% thought they would continue to do so after the COVID-19 emergency (S3 Table).

**Social contact.** Compared to before lockdown, respondents had less face-to-face contact with friends and family members during lockdown, but more regular telephone calls, video calls, and text or instant messages (S4 and S5 Figs). Over one-third (33.7%) had more contact with their neighbours during lockdown; 19.5% had less contact; of 101 who reported a change, 62.4% rated this change positively, 31.7% neutral, and 5.9% negatively (S4 Table).

**Self-reported physical and mental health and loneliness.** In total, 55.8% rated their physical health before lockdown as being either excellent or very good; this fell to 47.8% during lockdown (Fig 1). Before lockdown, 85.1% rated their emotional and mental health as being either excellent or very good; this fell to 68.6% during lockdown (Fig 2). Over one-third (36.5%) of respondents felt nervous or stressed because of COVID-19, and less than one quarter (23.8%) felt lonely during lockdown (S5 Table).

**Lifestyle factors.** Of 121 respondents who drink alcohol, 11.7% consumed more alcohol during lockdown; 24.2% consumed less. There were 75 (39.7%) ex-smokers, 2 (1.1%) current smokers, and 112 (59.3%) had never smoked. Few reported a change in diet during lockdown: 18.5% had a healthier diet; 7.9% had a less healthy diet; 10.1% were eating more; and 12.7% were eating less. Almost half of respondents (48.2%) reported doing less physical activity during lockdown, whereas 17.5% did more, and 34.4% did the same amount. Over half of respondents (62.6%) returned to an old pastime or started a new one during lockdown (S6 Table). Of 18 pastimes, the most popular were reading (65.3%), watching films or television (63.2%), and gardening (54.0%; Fig 3).

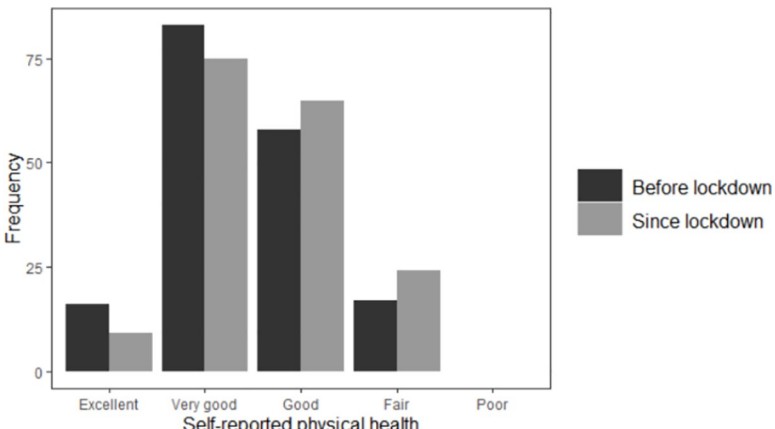

**Fig 1. Change in LBC1936 participants' self-reported physical health after COVID-19 measures introduced.**

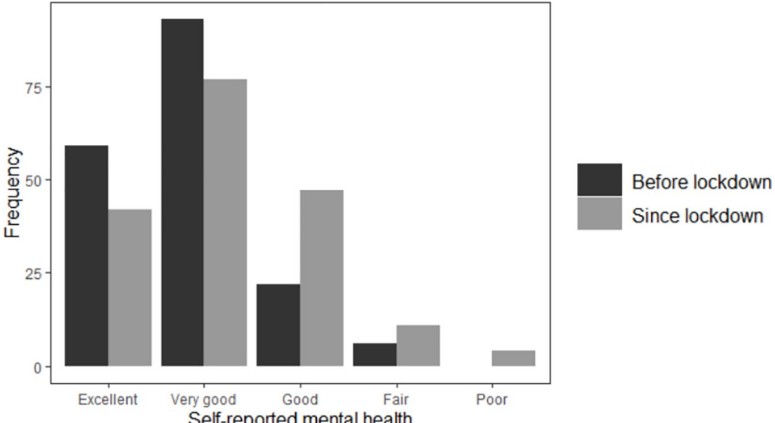

**Fig 2. Change in LBC1936 participants' self-reported emotional and mental health after COVID-19 measures introduced.**

## Correlations between characteristics at age 82 (or earlier) and COVID-19 outcomes at age 84

Spearman's rank correlations for the analytical sample ($n$ = 182) are presented in Table 3.

**Adherence to guidance.** Leaving home less frequently during lockdown was correlated with less professional occupational class, more chronic diseases, higher Townsend Disability Scale score, poorer self-rated general health, and lower gf at age-82.

**Impact on day-to-day living.** Using the internet more often during lockdown was correlated with being female, currently living alone, higher age-82 gf, and greater anxiety symptoms. Change in daily routine was correlated with not living alone and higher age-82 general health literacy. No variables correlated with getting additional help during lockdown.

**Self-reported physical and mental health and loneliness.** poorer self-reported physical health during lockdown was correlated with being older and male, lower gf, more chronic diseases, higher Townsend Disability Scale score, poorer self-rated general health, greater anxiety and depression symptoms, and lower emotional stability, conscientiousness, and extraversion at age-82. Poorer self-reported mental health during lockdown was correlated with currently living alone, more chronic diseases, poorer self-rated general health, greater anxiety and depression symptoms, and lower emotional stability, and extraversion at age-82. COVID-19

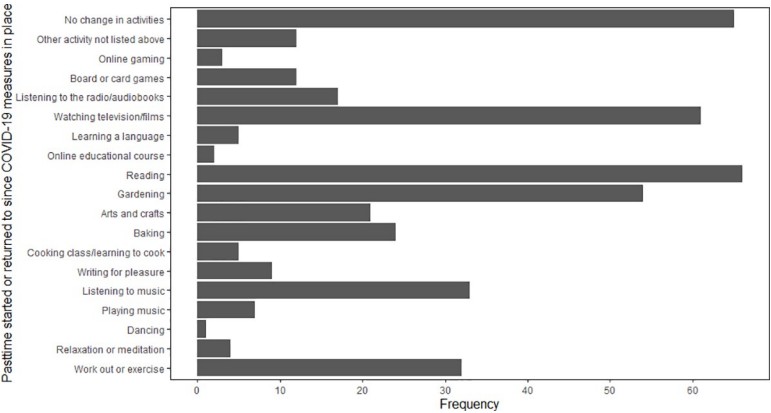

**Fig 3. Pastimes returned to or taken up during lockdown.**

**Table 3. Spearman's rank correlations between age-82 (or earlier) characteristics and COVID-19 questionnaire outcomes.**

| | Adherence to guidance | Impact on day-to-day living | | | Self-reported physical and mental health and loneliness | | | | Lifestyle | |
|---|---|---|---|---|---|---|---|---|---|---|
| | Frequency of leaving home | Change in internet usage | Gets additional help | Change in daily routine | Self-reported physical health | Self-reported mental health | COVID-19-related stress or nervousness | Loneliness | Decrease in physical activity | New pastime |
| Age[a] | -0.01 | 0.03 | 0.00 | 0.05 | 0.18* | 0.14 | -0.02 | 0.08 | 0.03 | 0.06 |
| Sex (*n* male) | -0.12 | -0.23** | 0.03 | -0.05 | -0.20** | 0.03 | 0.13 | 0.06 | 0.00 | -0.18* |
| **Demographics** | | | | | | | | | | |
| Adulthood occupational class | -0.18* | 0.09 | 0.14 | -0.15 | 0.10 | -0.03 | -0.01 | 0.11 | 0.22** | -0.02 |
| Living alone[b] (*n* yes) | -0.02 | 0.20* | -0.01 | 0.16* | 0.07 | -0.16* | -0.19* | -0.40*** | -0.03 | 0.14 |
| **Cognitive** | | | | | | | | | | |
| Fluid cognitive ability 'gf' | 0.19* | 0.16* | 0.02 | -0.13 | -0.27*** | 0.37 | 0.07 | -0.09 | -0.26*** | -0.09 |
| General health literacy | 0.13 | -0.17 | 0.01 | -0.20* | -0.02 | -0.10 | 0.07 | -0.03 | -0.13 | -0.16* |
| **Health** | | | | | | | | | | |
| Number of chronic diseases | -0.15* | 0.03 | 0.03 | 0.03 | 0.22** | 0.15* | 0.01 | 0.12 | 0.12 | 0.06 |
| Townsend disability scale score | -0.16* | -0.04 | 0.00 | 0.11 | 0.32*** | 0.14 | 0.05 | 0.17* | 0.07 | 0.05 |
| Self-rated general health | -0.27*** | -0.07 | -0.02 | 0.00 | 0.52*** | 0.32*** | 0.10 | 0.16* | 0.09 | 0.12 |
| **Mood** | | | | | | | | | | |
| Anxiety symptoms | -0.05 | -0.20** | 0.13 | -0.01 | 0.17* | 0.36*** | 0.28*** | 0.31*** | -0.01 | -0.09 |
| Depression symptoms | -0.07 | -0.01 | 0.09 | 0.13 | 0.33*** | 0.26*** | 0.12 | 0.11 | -0.03 | 0.09 |
| **Personality** | | | | | | | | | | |
| Emotional stability | 0.04 | 0.07 | -0.10 | -0.04 | -0.29*** | -0.43*** | -0.39*** | -0.29*** | 0.00 | 0.01 |
| Conscientiousness | 0.08 | 0.11 | -0.08 | 0.05 | -0.20** | -0.10 | -0.09 | -0.04 | -0.00 | -0.03 |
| Extraversion | 0.09 | -0.06 | 0.05 | -0.05 | -0.25** | -0.22** | -0.03 | -0.09 | -0.14 | -0.12 |

*$p < .05$

**$p < .01$

***$p < .001$; Independent variables are from age-82 unless otherwise stated. All p-values corrected using Holm-Bonferroni correction [71].

[a] Age is age in days at time of questionnaire (mean age 84).

[b] Living alone at time of questionnaire (mean age 84).

related stress or nervousness during lockdown was correlated with currently living alone, greater anxiety symptoms, and lower emotional stability at age-82. Feeling lonely during lockdown was correlated with living alone, higher Townsend Disability Scale score, poorer self-rated general health, greater anxiety symptoms, and lower emotional stability at age-82.

**Lifestyle.** Doing less physical activity during lockdown was correlated with having a less professional occupational class and lower age-82 gf. Returning to an old pastime or starting a new one during lockdown was correlated with being female and higher age-73 general health literacy.

## Regression analyses with age-82 (or earlier) characteristics as independent variables and age-84 COVID-19 questionnaire responses as outcomes

Results of final regression models for each outcome are displayed in Table 4; full results of all individual regression models are provided in S8–S16 Tables.

**Table 4. Odds ratios (95% confidence intervals) for final regression models of COVID-19 outcomes predicted by characteristics at age-82 (or earlier).**

| | Adherence to guidance | Impact on day-to-day living | | Self-reported physical and mental health and loneliness | | | | Lifestyle | |
|---|---|---|---|---|---|---|---|---|---|
| | Frequency of leaving home [c] | Change in internet usage [c] | Change in daily routine [d] | Self-reported physical health [d] | Self-reported mental health [d] | COVID-19-related stress or nervousness [c] | Loneliness [c] | Decrease in physical activity [d] | New pastime [c] |
| Age[a] | 0.92 (0.69–1.23) | 0.97 (0.68–1.38) | 1.16 (0.86–1.57) | 1.45 (1.04–2.04)* | 1.27 (0.94–1.73) | 0.94 (0.66–1.33) | 1.04 (0.68–1.62) | 1.13 (0.85–1.51) | 0.92 (0.66–1.28) |
| Sex    Male | Reference | Reference | Reference | Reference | Reference | Reference | Reference | Reference | Reference |
|     Female | 0.56 (0.30–1.02) | 2.32 (1.12–4.86)* | 1.08 (0.55–2.12) | 0.56 (0.28–1.11) | 1.11 (0.57–2.16) | 1.55 (0.73–3.31) | 0.48 (0.17–1.26) | 1.08 (0.61–1.89) | 1.89 (0.96–3.75) |
| **Demographics** | | | | | | | | | |
| Adulthood occupational class | 0.71 (0.51–0.98)* | - | - | - | - | - | - | 1.43 (1.04–1.96)* | - |
| Living alone[b]    Alone | Reference | Reference | Reference | Reference | Reference | Reference | Reference | Reference | Reference |
|     Not alone | - | 0.65 (0.38–1.10) | 1.83 (0.92–3.68) | - | 0.53 (0.28–1.02) | 0.65 (0.38–1.11) | 0.15 (0.07–0.31)***† | - | - |
| **Cognitive** | | | | | | | | | |
| Fluid cognitive ability 'gf' | 1.24 (0.88–1.74) | 1.53 (1.03–2.33)* | - | 0.73 (0.50–1.05) | - | - | - | 0.68 (0.49–0.93)* | - |
| General health literacy | - | - | 0.69 (0.46–1.01) | - | - | - | - | - | 1.36 (0.92–2.02) |
| **Health** | | | | | | | | | |
| Number of chronic diseases | 0.87 (0.60–1.26) | - | - | 0.98 (0.65–1.47) | 1.20 (0.82–1.75) | - | - | - | - |
| Townsend disability scale score | 0.73 (0.46–1.13) | - | - | 1.31 (0.74–2.37) | - | - | 1.67 (0.92–3.14) | - | - |
| Self-rated general health[c] | 0.62 (0.42–0.92)* | - | - | 3.99 (2.31–7.11)***† | 1.48 (0.99–2.24) | - | - | - | - |
|     Excellent | - | - | - | - | - | - | 0.17 (0.001–26.69) | - | - |
|     Very good | - | - | - | - | - | - | 0.64 (0.01–65.46) | - | - |
|     Good | - | - | - | - | - | - | 0.34 (0.003–33.48) | - | - |
|     Fair | - | - | - | - | - | - | 3.32 (0.03–399.82) | - | - |
|     Poor | Reference | Reference | Reference | Reference | Reference | Reference | Reference | Reference | Reference |
| **Mood** | | | | | | | | | |
| Anxiety symptoms | - | 1.31 (0.92–1.90) | - | 0.84 (0.54–1.30) | 1.15 (0.76–1.73) | 0.99 (0.63–1.55) | 1.76 (1.01–3.14)* | - | - |
| Depression symptoms | - | - | - | 1.17 (0.77–1.78) | 1.03 (0.71–1.50) | - | - | - | - |
| **Personality** | | | | | | | | | |
| Emotional stability | - | - | - | 0.81 (0.51–1.26) | 0.54 (0.35–0.81)**† | 0.40 (0.24–0.62)***† | 0.76 (0.45–1.24) | - | - |
| Conscientiousness | - | - | - | 0.83 (0.57–1.20) | - | - | - | - | - |
| Extraversion | - | - | - | 0.83 (0.58–1.17) | 0.89 (0.64–1.24) | - | - | - | - |

*$p < .05$

**$p < .01$

***$p < .001$; Independent variables are from age-82 unless otherwise stated

[a] age is age in days at time of questionnaire (mean age 84)

[b] living alone at time of questionnaire (mean age 84).

†associations remain significant after multiple testing correction via false discovery rate (FDR) estimation.

[c] models were binary logistic regression models.

[d] models were ordinal logistic regression models.

**Adherence to guidance.**   Leaving home less frequently during lockdown was associated with a less professional occupational class and poorer age-82 self-rated general health.

**Impact on day-to-day living.**   The odds of using the internet more during lockdown were greater for women and higher age-82 fluid cognitive ability. No measures were significantly associated with change in daily routine in the final model.

**Self-reported physical and mental health and loneliness.**   Odds of poorer self-reported physical health during lockdown were increased for those who were older, and had poorer age-82 self-rated general health. Odds of poorer self-reported emotional and mental health during lockdown were lower for those with higher emotional stability. Odds of COVID-19-related stress or nervousness during lockdown were lower for those with higher emotional stability trait scores. Odds of being lonely during lockdown were higher for those with greater age-82 anxiety symptoms and lower for those not living alone.

**Lifestyle.**   Decreased physical activity was associated with less professional occupational class, and lower general cognitive ability. There were no significant associations with participation in pastimes in fully adjusted models.

## Discussion

In a well-characterised sample of community-dwelling 84-year-olds from the LBC1936, we conducted a questionnaire examining the impact of Scottish COVID-19 lockdown guidance on the lives of older people. This is one of the largest studies–exclusively in adults aged over 80 years–of psychosocial factors, health and lifestyle in relation to COVID-19 to-date. This study offers an important snapshot of the impact on octogenarians following two months of stringent lockdown restrictions. By linking questionnaire responses during lockdown with characteristics measured at least two years earlier (age-82), it highlights possible risk and protective factors for health and behaviour during lockdown, and adds to what is known about effects of the COVID-19 lockdown on older people.

Reassuringly, our findings indicate that this group of older adults coped relatively well during lockdown. Respondents had little direct experience of the virus and mostly had good self-reported physical and mental health, but experienced some changes to their routines and activities. However, the lockdown experience was not universally positive. Some experienced modest declines in self-reported mental and physical health; one-third of respondents experienced COVID-19-related stress or nervousness, and 25% felt lonely during lockdown. Results of our regression analyses highlight individual differences that may be associated with increased risk of, or protection against, negative outcomes during the current and future waves of the pandemic.

Greater loneliness during lockdown was associated with living alone and greater age-82 anxiety symptoms. Evidence on mental health during lockdown is mixed; some studies suggest those over age 70 are less likely to feel stressed or anxious or report a negative effect on their mental health than younger age groups, and others report that odds of reporting high anxiety during COVID-19 was twice as likely in those over age 75 than those under 24 [7, 35]. Non-responders in the current study were more likely to be living alone and to have greater age-82 anxiety symptoms; therefore, our results might underestimate the proportion of older people experiencing loneliness and the magnitude of the associations between loneliness, living alone, and anxiety in the general population. Given the known negative consequences of loneliness for older adults [10–16], public health measures to counteract loneliness are likely to be increasingly important.

Individuals who adhered to guidance by leaving home less frequently during lockdown had poorer self-rated health and less professional occupational class. This may reflect that those

with previous health problems, and those who may face greater material disadvantages (such as having fewer financial resources due to past occupational status) when managing the stress of the virus [73], may take greater precautions to safeguard their health. Unlike previous research on adherence to guidelines during the pandemic, we found no association between adherence to this guideline and sex [74] or marital status [75]. Overall, results complement findings that the threat of COVID-19 is perceived to be lower for those who are healthier and have higher income [23], and is broadly in line with previous research showing associations between socio-economic factors such as level of education and pandemic guideline compliance [74–77].

Our finding that almost half of respondents reported decreased physical activity builds upon previous findings of lower levels of vigorous physical activity in adults during lockdown [37]. This may be particularly important in the context of previous studies showing associations between physical fitness and cognitive ability [78] and cognitive decline [18] in older age, and on the mediating effect of physical activity on the relationship between stress levels and mental health [79]. Healthcare providers considering web-based provision of information and interventions should consider that online campaigns may not reach all parts of the older adult population equally; men and those with lower age-82 gf were less likely to report increased internet usage during lockdown. A more positive outcome is that over 60% of respondents started or returned to a pastime during lockdown. A previous LBC1936 study found that playing analogue games was associated with less cognitive decline in those aged 70–79 [80], however cognitive benefits of different types of pastimes may vary across different age groups [78]. Future studies might examine whether there were benefits associated with taking part in specific types of pastime (e.g active versus passive) during lockdown.

## Strengths and limitations

This was one of the first studies to collect data on the impact of the COVID-19 lockdown on octogenarians. This adds to and strengthens the current COVID-19 literature, specifically in terms of examining outcomes in older adults. Few studies to date have included large samples of older adults; where older adults have been included, sample sizes tend to be low. Even among larger-scale studies, and those which sampled a wider range of older ages (e.g. extending from age 70 into late 80s), few report equivalent sample sizes to that achieved in the current study, with others ranging between only 22 participants over age 80 [26] to 237 when adults aged 60 years and below are included [27]. Additionally, LBC1936 members have a narrow age range, which reduces the likelihood that results are confounded by variation in age. Due to the wealth of previously collected data, the current study also had the rare advantage of being able to link COVID-19 questionnaire outcomes with longitudinal characteristics, thereby avoiding problems inherent in retrospective data collection, such as results being affected by poor recall memory or current circumstances. Furthermore, multivariate models were able to include relevant variables to minimise confounding. The questionnaire was distributed at an expedient time, when lockdown guidance was consistent for all respondents, and respondents completed it around two months after the onset of lockdown, so responses were unlikely to be affected by a short-lived peak in anxiety or emotional distress which might have occurred when the pandemic first took hold.

This study has limitations. The LBC1936 is a self-selecting sample consisting of mostly white Scottish participants who are likely to be healthier than the general population, and participants who have completed more waves of data collection have higher cognitive ability and physical fitness, and more professional occupational social class than those lost to attrition [44, 45]. Furthermore, as previously noted, responders to the LBC1936 COVID-19 questionnaire

tended to have a more professional occupational status, more years of education, and better physical and cognitive ability than LBC1936 participants who did not respond. Consequently, the results reported here are likely to be modest underestimations of the true effect size in the general population, as has been demonstrated previously in this cohort [81–84]. While reasons for lack of response are unknown, it is possible those who did not respond either did not have access to the necessary equipment to access the questionnaire online, or had lower computer literacy, further affecting generalisability of results. For example, if lower computer literacy is related to the outcomes under investigation in the current study, the results reported here are likely to be an underestimation of the true effect size in the population as a whole. It could also be considered a limitation that COVID-19 survey questions referred to a relatively long period of time from lockdown to survey completion, spanning 2 months, therefore responses may lack consistency if some participants respond while only reflecting on their most recent behaviour and experiences. However, this is likely to have been substantially mitigated by the fact that questions were explicitly worded in order to control for any variability in the period on which participants based their answers. In addition, Scottish Government guidance for those over age 80 did not change at any point while the LBC1936 COVID-19 questionnaire was live, therefore it in not possible that responses would have been confounded by any change in lockdown guidance. The questionnaire relied on self-report, without objective measures to gauge the accuracy of results. In addition, though we have interpreted results as being a consequence of the COVID-19 lockdown and associated guidance, participants were, on average, 2 years older than they were at their most recent assessment, therefore we cannot rule out the possibility that results were somewhat influenced by age-related physical and cognitive declines. Recently published analyses based on the same cohort as the present study have demonstrated that while there were small but significant changes in social support and physical activity variables between wave 5 (age 82) and the beginning of the COVID-19 pandemic (age 84), changes in social support and physical activity prior to the pandemic (between age 79 and 82), assumed to reflect ageing-related effects, were not significant [21]. As such, it is likely that results reported in the current study were not due solely to age-related change.

## Conclusions

In this study, we reported on the impact of COVID-19 lockdown in Scotland on psychosocial factors, health, and lifestyle in members of LBC1936. Results indicated that those with lower cognitive functioning, less professional occupational social class, lower emotional stability, greater anxiety symptoms, and living alone may be particularly at risk of negative lockdown-related outcomes, including loneliness and reduced physical activity, poorer self-reported mental and physical health, and greater stress and nervousness. Older adults with these characteristics may benefit from additional support to reduce the risk of negative outcomes. Additionally, policy makers and healthcare providers might focus on outcomes of loneliness and physical activity, which are widely known to have attendant negative consequences.

## Supporting information

**S1 Checklist. STROBE statement—checklist of items that should be included in reports of observational studies.**
(DOCX)

**S1 Fig. LBC1936 participants' (n = 4) responses to duration of COVID-19 symptoms.**
(DOCX)

**S2 Fig. Types of news source used by LBC1936 participants to stay informed about COVID-19.**
(DOCX)

**S3 Fig. News sources rated most helpful by LBC1936 participants staying informed about COVID-19.**
(DOCX)

**S4 Fig. LBC1936 participants' self-reported methods of contact with family before and during COVID-19 lockdown.**
(DOCX)

**S5 Fig. LBC1936 participants' self-reported methods of contact with friends before and during lockdown.**
(DOCX)

**S1 Table. LBC1936 participant responses to COVID-19 questionnaire: Experience of COVID-19.**
(DOCX)

**S2 Table. LBC1936 participant responses to COVID-19 questionnaire: COVID-19 knowledge and guidance.**
(DOCX)

**S3 Table. LBC1936 participant responses to COVID-19 questionnaire: Living situation and impact on day-to-day living.**
(DOCX)

**S4 Table. LBC1936 participant responses to COVID-19 questionnaire: Social connectedness.**
(DOCX)

**S5 Table. LBC1936 participant responses to COVID-19 questionnaire: Self-reported physical and mental health and loneliness.**
(DOCX)

**S6 Table. LBC1936 participant responses to COVID-19 questionnaire: Behaviour.**
(DOCX)

**S7 Table. Factor loadings and percentage of variance explained by first unrotated component from principal components analysis of general fluid ability and general health literacy items.**
(DOCX)

**S8 Table. Odds ratios (95% confidence intervals) for decreased frequency of leaving the home since COVID-19 lockdown.**
(DOCX)

**S9 Table. Odds ratios (95% confidence intervals) for increased internet usage since COVID-19 lockdown.**
(DOCX)

**S10 Table. Odds ratios (95% confidence intervals) for reporting a greater change in daily routine since COVID-19 lockdown.**
(DOCX)

**S11 Table. Odds ratios (95% confidence intervals) for reporting poorer self-reported physical health since COVID-19 lockdown measures introduced.**
(DOCX)

**S12 Table. Odds ratios (95% confidence intervals) for reporting poorer self-reported mental health since COVID-19 lockdown measures introduced.**
(DOCX)

**S13 Table. Odds ratios (95% confidence intervals) for experiencing COVID-19 related stress or nervousness during COVID-19 lockdown.**
(DOCX)

**S14 Table. Odds ratios (95% confidence intervals) for experiencing loneliness during COVID-19 lockdown.**
(DOCX)

**S15 Table. Odds ratios (95% confidence intervals) for a decrease in physical activity since COVID-19 lockdown measures introduced.**
(DOCX)

**S16 Table. Odds ratios (95% confidence intervals) for returning to or starting a new pastime since COVID-19 lockdown measures introduced.**
(DOCX)

**S1 Appendix.**
(DOCX)

## Acknowledgments

We are grateful to all participants of the Lothian Birth Cohort 1936 study. We thank the LBC1936 research team, and the research nurses at the Wellcome Trust Clinical Research Facility, Western General Hospital, Edinburgh, for their contributions to previous waves of the study. We thank Professor Ian Deary for suggestions on study design and on earlier versions of this paper.

## Author Contributions

**Conceptualization:** Adele M. Taylor, Danielle Page, Judith A. Okely, Janie Corley, Miles Welstead, Paul Redmond, Tom C. Russ, Simon R. Cox.

**Data curation:** Adele M. Taylor, Danielle Page, Paul Redmond.

**Formal analysis:** Adele M. Taylor, Danielle Page.

**Funding acquisition:** Adele M. Taylor, Tom C. Russ, Simon R. Cox.

**Investigation:** Adele M. Taylor, Danielle Page, Judith A. Okely, Janie Corley, Miles Welstead, Tom C. Russ, Simon R. Cox.

**Methodology:** Adele M. Taylor, Danielle Page, Judith A. Okely, Janie Corley, Miles Welstead.

**Project administration:** Adele M. Taylor, Danielle Page.

**Software:** Paul Redmond.

**Supervision:** Tom C. Russ, Simon R. Cox.

**Validation:** Adele M. Taylor, Danielle Page.

**Visualization:** Adele M. Taylor, Danielle Page.

**Writing – original draft:** Adele M. Taylor, Danielle Page.

**Writing – review & editing:** Adele M. Taylor, Danielle Page, Judith A. Okely, Janie Corley, Miles Welstead, Barbora Skarabela, Paul Redmond, Tom C. Russ, Simon R. Cox.

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
