## [Decision Letter · Decision Letter 0]

24 Mar 2021

PONE-D-21-00465

Impact of COVID-19 lockdown on psychosocial factors, health, and lifestyle in Scottish octogenarians: the Lothian Birth Cohort 1936 Study

PLOS ONE

Dear Dr. Page,

Thank you for submitting your manuscript to PLOS ONE. After careful consideration, we feel that it has merit but does not fully meet PLOS ONE’s publication criteria as it currently stands. Therefore, we invite you to submit a revised version of the manuscript that addresses the points raised during the review process.

We look forward to receiving your revised manuscript.

Kind regards,

Muhammed Elhadi, MBBCh

Academic Editor

PLOS ONE

Additional Editor Comments:

The reviewers raised concerns about the transparency of data and also suggested to make the results section more readable.

Journal Requirements:

2. In the Methods section, please provide additional details regarding how the capacity to give informed consent was determined.

3. Please provide details regarding how the subset of participants from the Lothian Birth Cohort 1936 were selected for the current study. Please detail any inclusion and exclusion criterias applied.

6. Please include captions for *all* your Supporting Information files at the end of your manuscript, and update any in-text citations to match accordingly. Please see our Supporting Information guidelines for more information: http://journals.plos.org/plosone/s/supporting-information.

Reviewers' comments:

Reviewer's Responses to Questions

**Comments to the Author**

1. Is the manuscript technically sound, and do the data support the conclusions?

Reviewer #1: Yes

Reviewer #2: Yes

Reviewer #3: Yes

Reviewer #4: Yes

2. Has the statistical analysis been performed appropriately and rigorously? 

Reviewer #1: Yes

Reviewer #2: Yes

Reviewer #3: Yes

Reviewer #4: Yes

3. Have the authors made all data underlying the findings in their manuscript fully available?

Reviewer #1: Yes

Reviewer #2: Yes

Reviewer #3: Yes

Reviewer #4: Yes

4. Is the manuscript presented in an intelligible fashion and written in standard English?

Reviewer #1: Yes

Reviewer #2: Yes

Reviewer #3: Yes

Reviewer #4: Yes

5. Review Comments to the Author

Reviewer #1: Thank you for the opportunity to review this manuscript. Authors examine the impact of COVID-19 on octogenarians. The manuscript is well presented and investigates the effects of the pandemic on an underrepresented population in the current researches. While the ms offers useful insights on this matter, there are some issues that need to be addressed.

1. Regarding the choice of covariates. Authors state that “Measures […] were selected a priori based on previous associations between these variables and psychosocial factors, health and lifestyle in the LBC1936 cohort”. Authors should elaborate on this point, ideally also referencing the appropriate literature.

2. What was the rationale behind choosing not to administer the anxiety/depression scale used in wave5 but only using the previous scores as a covariate? The issue I see here is that the comparison pre/during pandemic is only made on a self-report assessment based on a few questions. It would have been interesting to see the results of such comparison on a validated instrument: this could have offered a better estimate of the actual impact of the pandemic on the sample’s mental health.

3. Discussion. Albeit, as suggested by Authors, studies in the general population may underrepresent octogenarians, a comparison with the existing literature could offer a more detailed interpretation of their results. To do so, a more thorough review of the literature is required. For instance, regarding adherence with protective measures, older aged groups are usually associated with higher compliance (e.g. Roma et al. 2020. How to improve compliance with protective health measures during the COVID-19 outbreak: testing a moderated mediation model and machine learning algorithms”. Int. J. Environ. Res. Public Health. https://doi.org/10.3390/ijerph17197252), and it would be interesting to have insight on what factors are associated with higher levels of compliance and how these findings relate to the literature regarding the general population. In other words, Authors should be more detailed in interpreting all of their results.

4. Discussion pag.22 line 398. “associated with living alone and greater age-82 anxiety symptoms” there seems to be a mistake here.

5. Strengths and Limitations. Regarding the online formats and the consequent limitation of available respondents. For clarity, Authors should elaborate on this point, because it likely had a significant impact on their results.

Reviewer #2: 1) In methods criteria of inclusion of participant should be defined as well as type of sampling

2) Please, I suggest to cite this papers on the relationship between psychosocial risk factors and physical health: 1) Chawla S, Kocher M. Physical activity at home during the COVID-19 lockdown in India: Need of the hour for optimum physical health and psychological resilience. J Health Soc Sci. 2020. 5(2): 187-192. 2) Chirico F, Heponiemi T, Pavlova M, Zaffina S, Magnavita N. Psychosocial Risk Prevention in a Global Occupational Health Perspective. A Descriptive Analysis. Int J Environ Res Public Health. 2019;16(14):2470. Published 2019 Jul 11. doi:10.3390/ijerph16142470

Reviewer #3: Title: Impact of COVID-19 lockdown on psychosocial factors, health, and lifestyle in Scottish octogenarians: the Lothian Birth Cohort 1936 Study

Summary:

This is a cross-sectional survey conducted in a subset of Lothian Birth Cohort (LBC) 1936 Study, aiming to examine the effects of the Scottish COVID-19 lockdown on psychosocial factors, health, and lifestyle among Scottish octogenarians. A total of 190 participants with a mean age of 84 years, responded to the online questionnaire survey in May 2020. The authors concluded that individual cognitive function, occupational class, self-rated health, anxiety, emotional stability, and living alone may be related to risk of lockdown-related psychosocial and physical outcomes. The manuscript is overall well-written.

Strengths:

1. A unique population: to date, there is limited COVID-19 literature on the impact of lock down on older adults in community-dwelling. This study offers a snapshot of two months lock down on Scottish octogenarians.

2. Leveraging the Lothian Birth Cohort: Its key advantage is that it has a well-defined population with rich data on cognitive ability, demographics, psychosocial, and health factors previously collected at age 73 (2007-2010, n=866), 76 (2011-2013, n=697), 79 (2014-2017, n=550), and 82 (2017-2019, n=431).

Comments:

1. It is helpful to include S8 table as a main table so the readers can have a good picture of the study participants characteristics. It would be helpful to include total sample size for respondents and non-respondents.

2. How representative of this subsample as compared the Scottish older people in general?

3. How to separate natural physical and cognitive decline due to aging from COVID-19 impact?

Reviewer #4: Great opportunity to understand the impact of COVID-19 on this older population. I think this is an important contribution to the literature but some work needs to help the readers easily digest the material.

Overall, I think the authors need to be a bit more careful in their statements. I think the intention vs the actual words may not match perfectly. I’ve noted places where I was concerned about the statement being bolder than the authors meant it to be. There was too much vagueness on ‘older’ population and how this was defined. Many statement included the word older adults and that could be anyone age 50 and older…but since it was not specified, super unclear. The methods section needs to be more clearly presented as does the results section.

Abstract, please add place (I’m sorry I do not know where Lothian Birth Cohort participants live).

“The physical, psychological, and social effects of coronavirus disease 2019 (COVID-19) are unprecedented.” Line 51

This is a bit of an overstatement. Historically, there were other epidemics/pandemics that had incredible impact, we just did not have the scientific and communication systems in place to document as well. This is unprecedented in our lifetime only and has impacted developed countries.

“We are yet to discover the effects of COVID-19 lockdown measures, particularly on older people” line 56 There is actually papers out that discuss the effect; it all depends on how ‘older people’ are defined. I suggest you make this more specific—particularly on people over 80 years of age and this when then be more accurate.

“have therefore endured some of the greatest restrictions for the 58 longest period.” Line 57. I would suggest placing this in context of UK? Or Scotland? This has not been true for all counties.

”at highest risk of severe illness” line 62. In the highest risk of severe illness category, right? Not the highest risk and your citation is on mortality which one could argue is severe illness but often this is being used to mean ICU admission, ECMO, ventilation… Either different citation or change to mortality and adjust the statement accordingly.

A prospective cohort study in UK acute care hospitals found the highest proportion of hospitalisations and mortality among those aged 80 and over (4). (line 64). What was the comparison group? I suspect other ages, right? Not those with HIV or multiple comorbidities or…? Need to add the comparison group.

There is a bit of a mishmash between severity (hospitalization) vs mortality in the intro. It would be cleaner to put the concepts together more logically.

“Because of this increased risk, those most vulnerable to the virus when lockdown began were asked to ‘shield’, remaining at home and strictly avoiding social contact with anyone outside

of their homes for at least 12 weeks.” Line 70. Again need place since this is a global sentence.

As you use vulnerable to describe at risk population, please define vulnerable, particularly the age at which someone is considered vulnerable. All you say is older people which I do not know what that means precisely. Similarly, the use of ‘older people’ is just vague and needs clarity. I would suggest defining it to fit with your study. “Data from older people during the pandemic are surprisingly limited. Studies of this age group are under represented in COVID-19 literature to date, “ (line 85) This entirely begs the question of ‘this age group”

”Data from older people during the pandemic are surprisingly limited.”line 85 Why is this surprising? Very few surveys have sufficient numbers of people in this age range.

“In studies which do include older adults, they often account for only 2-27% of the overall sample (19–22), with results based on fewer than 50 older adults in some cases” line 90 This does not support your argument that there are limited number of participants in a study if ¼ are older and ‘some case” but presumably not all have less than 50. Need to rephrase this whole argument. Recommend you keep it focused on your contribution of over 80 year olds. That is good enough. You don’t need a ton to justify this study.

There is also a fair amount of jumping between UK and Scotland. Why say UK lockdown since your population, I presume was Scotland. Lockdown in Scotland?

Please provide url for reports since the citations don’t actually work to get to the reports.

Individuals aged over 75 were more than twice as likely to report high anxiety during lockdown compared to those under 24 (31). Line 98. Why the comparison with very young adults? Kind of jarring. I looked up this report and I couldn’t find the information so don’t know who it was reported (such as by age group).

mostly middle aged and older adults” line 101 please define, age xxx

To date, they have attended four further

152 waves at mean ages 73 (2007-2010, n=866), 76 (2011-2013, n=697), 79 (2014-2017, n=550), and 82

153 (2017-2019, n=431). Probably not necessary to provide mean age; more important is why there is apprx 50% retention—dying or unable to participant to addresses the potential bias of these data. And 25% of these completing the survey. . For example does this sample basically represent ‘healthy 80+ year olds with high computer literacy? It is OK, but maybe a limitation of the study and I would suspect the results would be even stronger with less healthy folks included but we just don’t know

I’m not a fan of having the questionnaire listed in supplemental. No indication of validity or reliability of the questions; much too vague to me; as this is online, not sure why this isn’t included in some detail in the measure section. It seems like the questions were included in the table but it is not clear. What was each instrument? Most of these seem very genera and beg the question of validity without it being specified. A table that lists the different instruments and their validity/reliability would strengthen the paper; or if these have not been tested, then citations that used these questions.

It seems that sometime in May surveys were sent and potentially completed. The survey questions ask about the previous 6-10 weeks, last week in March, all of April, and then sometime in May. Most of the questions assume either average over this period or more likely people will respond to their behavior in the past week or two. This is a limitation to the study since there could very well be a lack of consistency by respondents in thinking about this time period as well as answering it accurately for this time period.

Collected at wave 1 (mean age 70). For sex row; age not necessary.

Question collected previously need to have more clear description. Long list of cognitive ability and just using citations leave much work to the readers to sort this out.

Was BMI components measured or self-reported?

The table was very difficult to process. It would be easier if broken up between the 3 purposes (*, a, b)

“10 COVID-19 questionnaire outcomes” line 206. What is this. Super difficult to follow. Of these correlated variables, were they all considered to be stable and not have any meaningful change from time of collection to current?

“grouped into blocks by variable type” no idea what this means

“between ‘before’ and ‘during’ lockdown ratings for self-reported physical and mental health” line 220 not clear what questions were used for this? Validated questions or not?

The long correlation between age 82 and 84 was way too much text; a table is sufficient; similarly, there is considerable redundancy between the text and table 3. Both are unnecessary. If the text had less numbers in it, it would read easier, Lump those that go in the same direction and refer to the table for specific odds.

To me what was missing was looking at gender. So stratifying by gender since the risk for severe COVID infection was so much higher for men vs women; did gender influence the outcomes. I would also imagine living alone and male vs female would also be a very different story w/r to impact of COVID but don’t know if you are sufficiently powered.

On octogenarians” line 384. Is this true, they were all in their 80s? If so that is a great way to describe this population from the beginning. This was not obvious until the discussion and it should have been part of the intro…

“which may be underestimates of the experiences in the general population.” Line 447. This will never generalize to the general population; but it could generalize to octogenarians who face lock down but more likely generalizes to this study since there were so many significant differences between those who completed this study and those who did not, as noted in the results section.

6. PLOS authors have the option to publish the peer review history of their article (what does this mean?). If published, this will include your full peer review and any attached files.

Reviewer #1: No

Reviewer #2: No

Reviewer #3: No

Reviewer #4: No

---

## [Author Response · Author response to Decision Letter 0]

11 May 2021

We thank the Editor and Reviewers for the opportunity to revise and resubmit our manuscript, “Impact of COVID-19 lockdown on psychosocial factors, health, and lifestyle in Scottish octogenarians: the Lothian Birth Cohort 1936 Study” to PLOS ONE. We are grateful to Dr Elhadi and the Reviewers for their positive comments, and suggested changes which have allowed us to improve the quality of our manuscript. In response, we have made several major changes, which are explained in full in our Cover Letter and Response to Reviewers documents. We hope that these changes are sufficient to make our manuscript suitable for publication in PLOS ONE, and very much look forward to hearing from you.

---

## [Decision Letter · Decision Letter 1]

31 May 2021

Impact of COVID-19 lockdown on psychosocial factors, health, and lifestyle in Scottish octogenarians: the Lothian Birth Cohort 1936 Study

PONE-D-21-00465R1

Dear Dr. Page,

We’re pleased to inform you that your manuscript has been judged scientifically suitable for publication and will be formally accepted for publication once it meets all outstanding technical requirements.

Kind regards,

Muhammed Elhadi, MBBCh

Academic Editor

PLOS ONE

Additional Editor Comments (optional):

Reviewers' comments:

Reviewer's Responses to Questions

**Comments to the Author**

1. If the authors have adequately addressed your comments raised in a previous round of review and you feel that this manuscript is now acceptable for publication, you may indicate that here to bypass the “Comments to the Author” section, enter your conflict of interest statement in the “Confidential to Editor” section, and submit your "Accept" recommendation.

Reviewer #1: All comments have been addressed

Reviewer #2: All comments have been addressed

2. Is the manuscript technically sound, and do the data support the conclusions?

Reviewer #1: Yes

Reviewer #2: Yes

3. Has the statistical analysis been performed appropriately and rigorously? 

Reviewer #1: Yes

Reviewer #2: Yes

4. Have the authors made all data underlying the findings in their manuscript fully available?

Reviewer #1: (No Response)

Reviewer #2: Yes

5. Is the manuscript presented in an intelligible fashion and written in standard English?

Reviewer #1: Yes

Reviewer #2: Yes

6. Review Comments to the Author

Reviewer #1: (No Response)

Reviewer #2: The manuscript is original and a good fit to this journal.

Thank you for addressing all my comments

Congratulations for your work!

7. PLOS authors have the option to publish the peer review history of their article (what does this mean?). If published, this will include your full peer review and any attached files.

Reviewer #1: No

Reviewer #2: No

---

## [Editor Report · Acceptance letter]

8 Jun 2021

PONE-D-21-00465R1 

Impact of COVID-19 lockdown on psychosocial factors, health, and lifestyle in Scottish octogenarians: the Lothian Birth Cohort 1936 Study 

Dear Dr. Page:

I'm pleased to inform you that your manuscript has been deemed suitable for publication in PLOS ONE. Congratulations! Your manuscript is now with our production department. 

Kind regards, 

on behalf of

Dr. Muhammed Elhadi 

Academic Editor

PLOS ONE